# Self-Replication of Prion Protein Fragment 89-230 Amyloid Fibrils Accelerated by Prion Protein Fragment 107-143 Aggregates

**DOI:** 10.3390/ijms21197410

**Published:** 2020-10-08

**Authors:** Tomas Sneideris, Mantas Ziaunys, Brett K.-Y. Chu, Rita P.-Y. Chen, Vytautas Smirnovas

**Affiliations:** 1Institute of Biothechnology, Life Sciences Center, Vilnius University, LT-10257 Vilnius, Lithuania; sneideris.t@gmail.com (T.S.); mantas.ziaunys@gmail.com (M.Z.); 2Institute of Biological Chemistry, Academia Sinica, Taipei 115, Taiwan; brettchu007@gmail.com (B.K.-Y.C.); pyc@gate.sinica.edu.tw (R.P.-Y.C.); 3Department of Chemistry, National Taiwan University, Taipei 106, Taiwan; 4Institute of Biochemical Sciences, National Taiwan University, Taipei 106, Taiwan

**Keywords:** amyloid, prion, aggregation, self-replication

## Abstract

Prion protein amyloid aggregates are associated with infectious neurodegenerative diseases, known as transmissible spongiform encephalopathies. Self-replication of amyloid structures by refolding of native protein molecules is the probable mechanism of disease transmission. Amyloid fibril formation and self-replication can be affected by many different factors, including other amyloid proteins and peptides. Mouse prion protein fragments 107-143 (PrP(107-143)) and 89-230 (PrP(89-230)) can form amyloid fibrils. β-sheet core in PrP(89-230) amyloid fibrils is limited to residues ∼160–220 with unstructured N-terminus. We employed chemical kinetics tools, atomic force microscopy and Fourier-transform infrared spectroscopy, to investigate the effects of mouse prion protein fragment 107-143 fibrils on the aggregation of PrP(89-230). The data suggest that amyloid aggregates of a short prion-derived peptide are not able to seed PrP(89-230) aggregation; however, they accelerate the self-replication of PrP(89-230) amyloid fibrils. We conclude that PrP(107-143) fibrils could facilitate the self-replication of PrP(89-230) amyloid fibrils in several possible ways, and that this process deserves more attention as it may play an important role in amyloid propagation.

## 1. Introduction

Several neurodegenerative human health disorders, such as Alzheimer’s disease (AD) [1], Parkinson’s disease (PD) [2], as well as prion diseases [3] are all closely linked to a process, where particular proteins fail to maintain their native conformational state and form fibrillar amyloid aggregates possessing a cross-β structure. It is believed that the latter structure arises from arrays of β-sheets running parallel to the long axis of the fibril, while β-strands in an individual sheet are arranged perpendicularly to the fibril’s axis [4,5].

Understanding the process of such aggregate formation is of utmost importance, since amyloid-related disorders are becoming some of the most common medical conditions in the aging society [6], with more than 50 million people worldwide afflicted by AD and PD [1,2,7]. Currently, there are no cures or disease-modifying drugs available for most of these disorders, and even compounds that show promising results in vitro experience very high failure rates during clinical trials [4,8,9,10]. Analysis of such failures revealed that the main factors preventing successful development of effective anti-amyloid drugs are the relatively poor understanding of amyloid aggregation mechanisms; the lack of knowledge of the specific species and aggregation step(s) that may be affected by the molecule in question; the lack of methods to monitor the aggregation reaction in a reliable manner [4,11].

Prion diseases stand out among other amyloid-related disorders due to the possible transmission between individuals and, in some cases, even between species [11]. The ability of protein amyloid aggregates to self-replicate by recruiting and refolding native protein molecules is one of the driving forces for spreading the disease [11]. The recent decade came up with many studies shedding light on the molecular-level events underlying amyloid formation. Currently, there are two defined events related with amyloid fibril self-replication: fibril elongation and fibril surface-catalyzed nucleation (often referred to as secondary nucleation) [4,12,13,14,15]. Fibril elongation is the main event that contributes to the growth of aggregate mass via the addition of individual monomers to nuclei/fibril ends, whereas secondary nucleation facilitates the proliferation of amyloid aggregates through the fibril-surface-induced formation of new growth competent nuclei. Elongating fibrils usually replicate the structure of the initial seed (with only a couple of exceptions of possible conformational switching reported [16,17]); however, there is increasing evidence suggesting that the structure of amyloid fibrils replicated via secondary nucleation route is dependent on the environment rather than on the template of seeds [18,19,20,21].

The structure of recombinant prion protein (rPrP) amyloid fibrils was extensively studied by different techniques all giving similar conclusions—parallel in-register β-sheet in the C-terminal region (starting from residues 160–170, up to residues 220–225) and a disordered N-terminus [22,23,24,25]. However, such amyloid fibrils were not infective in vivo. Studies of H/D exchange on brain-derived pathogenic prion isoform (PrP${}^{Sc}$) showed similar highly packed structure as in case of rPrP amyloid-like fibrils but in a much longer region (entire 90–230 region) [26]. This suggests that the structure of rPrP amyloid-like fibrils may be similar to the one of brain-derived PrP${}^{Sc}$, however, with a substantially shorter β-sheet core region. Recently it was demonstrated that amyloid-like fibrils generated from N-terminal PrP fragment 23-144 can be infectious in vivo [27]. These fibrils contain a parallel in-register β-sheet core in the region between ∼110–140 [28], suggesting a particular importance of this region for prion infectivity. Another study showed that synthetic mouse prion protein fragment consisting of 107–143 residues (PrP(107-143)) can form amyloid fibrils, which were found to catalyze fibrillization of full-length mouse prion protein (PrP(23-230)) [29]. We decided to study the mechanism of the PrP(107-143)-fibril-induced aggregation of mouse prion protein C-terminal fragment 89-230 (PrP(89-230)) expecting the reaction to proceed via one of the three possible ways. First, PrP(107-143) fibrils may elongate by self-replicating their structure. Second, the PrP(107-143) fibrils could start to elongate, however, upon recruiting longer proteins the β-sheet core would extend towards C-terminus within residues ∼107–225. Finally, the PrP(107-143) fibrils may be incapable of elongating; however, they could accelerate PrP(89-230) fibril formation without self-replication of their structure via heterogeneous fibril-surface-induced nucleation. Surprisingly, it seems that none of the above happens and PrP(107-143) fibrils are unable to initiate the de novo aggregation reaction of PrP(89-230); however, in combination with PrP(89-230) fibrils, a synergistic effect, resulting in enhanced seed-induced PrP(89-230) aggregation reaction, can be observed.

## 2. Results

First, we generated PrP(89-230) fibrils (Appendix A; Figure A1a) and PrP(107-143)-fibril-induced PrP(89-230) fibrils (Appendix A; Figure A1b) by continuous agitation of PrP(89-230) monomer without and with PrP(107-143) fibrils, respectively, at 37 ${}^{\circ }$C in 1 × PBS (pH 7.4) solution containing 3 M Urea and 1 M GuHCl; as described previously [29], and determined their secondary structure profiles using FTIR spectroscopy. The FTIR spectra of spontaneously formed PrP(89-230) fibrils and the PrP(107-143)-fibril-induced PrP(89-230) fibrils were almost identical (Appendix A; Figure A1c). Moreover, PK-digestion assay revealed that there are no evident variations in the size of the PK-resistant core between PrP(89-230) fibrils and PrP(107-143)-fibril-induced PrP(89-230) fibrils (Appendix A; Figure A1d). The results suggest that PrP(107-143) fibrils can not self-propagate their structure. In the aforementioned study [29], PrP(107-143)-fibril-induced aggregation reaction was performed under vigorous agitation conditions. It is known that mechanical agitation induces the fragmentation of existent fibrils and facilitates the nucleation step of amyloid fibril formation as well as the detachment of species from the air–water or solid–water interface, where proteins have a strong tendency to accumulate and where, in many cases, the nucleation step is likely to occur [10,15,30,31,32,33]. Therefore, agitation could favor nucleation, and hence the spontaneous formation of PrP(89-230) fibrils in our sample. In order to minimize possible events of nucleation and to favor fibril elongation, the PrP(107-143)-fibril-induced aggregation reaction was performed under quiescent conditions. Since, the aggregation reaction did not occur within reasonable experimental time (Appendix A
Figure A2), the experimental conditions were modified to obtain rapid fibril elongation kinetics (by increasing temperature to 60 ${}^{\circ }$C and decreasing denaturant concentration to 0.5 M of GuHCl) [34].

### 2.1. Aggregation Kinetics

Under quiescent conditions, in the absence of any preformed aggregates or in the presence of 10% (hereafter, percentage of aggregates is relative to total protein weight in solution) of PrP(107-143) fibrils, aggregation of monomeric PrP(89-230) did not occur within the experimental time (Figure 1a,b). PrP(89-230) aggregation reaction, induced by the addition of PrP(89-230) fibrils, shows a typical fibril-concentration-dependent change in the lag phase (Figure 1a,c), yielding sigmoidal shaped curves. If we compare the lag time ($t_{lag}$) values of seed-induced aggregation reaction performed in the absence and presence of 10% of PrP(107-143) fibrils, it appears that the lag phase in the presence of PrP(107-143) aggregates is shorter as if there was a higher concentration of PrP(89-230) seed present (Figure 1c). This divergence between $t_{lag}$ values becomes more evident as the concentration of PrP(89-230) seeds increases (Figure 1d). PrP(107-143) fibrils shorten the lag phase of the PrP(89-230) seed-induced aggregation reaction in a concentration-dependent manner (Figure 1e,f).

Additionally, we performed experiments where we introduced PrP(107-143) fibrils at different time-points of PrP(89-230) self-replication reaction (Appendix A; Figure A3). The introduction of PrP(107-143) fibrils at any time-point resulted in the shortening of $t_{lag}$; however, the strongest effect on $t_{lag}$ was most prominent when the PrP(107-143) fibrils were introduced at the very beginning of the reaction (Figure A3). Similarly, the introduction of the PrP(89-230) seed at any time-point of the aggregation reaction of monomeric PrP(89-230) performed in the presence of PrP(107-143) fibrils resulted in the shortening of $t_{lag}$ of PrP(89-230) aggregation reaction, with the strongest effect evident upon introduction of PrP(89-230) fibrils at the early stages of the aggregation reaction (Figure A4). Similarly, as described previously, PrP(107-143) fibrils alone had no evident effect on the aggregation reaction of monomeric PrP(89-230) within the experimental time.

### 2.2. Morphology

Sample analysis via atomic force microscopy (AFM) revealed that, while the PrP(107-143) does spontaneously form fibrils itself (Figure 2a), it seems that the PrP(107-143) fibrils do not induce the aggregation of monomeric PrP(89-230), as the fibrils observed after 1000 min of incubation (Figure 2e) are of similar length (∼200–600 nm) as the seed (Figure 2c). In the case when the PrP(89-230) fibrils (Figure 2d) were used as a seed, both in the presence or absence of PrP(107-143) aggregates, the resulting fibrils (Figure 2f,g) appear to be of similar length (several hundred of nm to few μm) to the spontaneously formed PrP(89-230) fibrils (Figure 2b). Please, also see Appendix A; Figure A5.

### 2.3. Secondary Structure

The Fourier-transform infrared (FTIR) spectroscopy spectra of all examined PrP(89-230) fibril samples possess similar structural profiles in the amide I/I’ region. PrP(89-230) fibril spectra exhibit maxima at ∼1627 cm${}^{-1}$ (with the main minima of the second derivative at ∼1622 cm${}^{-1}$ and a weaker one at ∼1628 cm${}^{-1}$) and a shoulder which is reflected by the minimum of the second derivative at ∼1662 cm${}^{-1}$ (Figure 3). The spectral profile of spontaneously formed PrP(107-143) fibrils is slightly different from the rest. The FTIR spectrum of latter fibrils exhibits a maximum at ∼1626 cm${}^{-1}$ (with the single minimum of the second derivative at ∼1626 cm${}^{-1}$). The results suggest that PrP(107-143) fibrils are unable to self-propagate by imprinting their structural template on PrP(89-230) monomers.

## 3. Discussion

Surprisingly, the addition of high amounts (10%) of PrP(107-143) fibrils into PrP(89-230) monomer solution had no effect on aggregation kinetics within the experimental time, whereas addition of even very low amounts (0.1%) of PrP(89-230) fibrils resulted in a substantial acceleration of aggregation reaction (Figure 1a,b). This suggests that heterogeneous seeding is extremely inefficient in this case. Interestingly, when both types of aggregates were co-added into PrP(89-230) monomer solution, the aggregation promoting effect was stronger than in case of the PrP(89-230)-seed alone. This raises the question: What is the origin of such an effect? From the $t_{lag}$ plot, it is evident that in the presence of 10% of PrP(107-143) fibrils $t_{lag}$ values of PrP(89-230) seed-induced aggregation reaction are lower as if the initial concentration of preformed PrP(89-230) aggregates was higher (Figure 1c and Appendix A
Figure A6). For instance, in the presence of 10% of PrP(107-143) fibrils and 2% of PrP(89-230) seed, the $t_{lag}$ value is ∼2.3 times lower, when compared to control, as if the initial concentration of PrP(89-230) seed was ∼3.9% (dashed lines in Appendix A
Figure A6). This suggests that 10% of PrP(107-143) fibrils should act as ∼1.9% of PrP(89-230) seed, which would point towards a cumulative aggregation promoting effect. If this was true, the aggregation promoting effect of PrP(107-143) fibrils alone should be evident; however, it is not, suggesting that the presence of PrP(89-230) seed is essential for PrP(107-143) fibrils to have an effect on the PrP(89-230) aggregation reaction. This implies that the aggregation promoting effect is synergistic in origin.

As suggested by AFM images (Figure 2e), it seems that the PrP(107-143) fibrils are incapable of elongatingthrough recruitment of PrP(89-230) monomers (or the elongation rate is extremely slow). Moreover, FTIR results suggest that the presence of PrP(107-143) fibrils has no effect on the structural profiles of the resulting PrP(89-230) fibrils. Thus, suggesting that the presence of PrP(107-143) fibrils affect only the lag phase of the PrP(89-230)-seed-induced aggregation reaction. The presence of monomeric PrP(107-143) has no effect on the aggregation kinetics of PrP(89-230)-seed-induced aggregation reaction (Appendix A; Figure A7), meaning that the synergistic effect most likely results from the interplay between fibrils formed by PrP(107-143) and PrP(89-230).

Primary nuclei capable of seeding aggregation reaction of monomeric proteins form relatively early during the lag phase of aggregation reaction [35], and the agitation facilitates this process. Therefore, it is likely that the aggregation-promoting effect of PrP(107-143) fibrils observed under agitated conditions [29] results in fact due to the interplay between PrP(107-143)-fibrils and the agitation-induced spontaneously formed aggregation prone PrP species.

The surface of preformed aggregates can catalyze formation of new growth-competent nuclei in three main ways. First, the protein monomers can condense on the surface of existing fibrils, which results in the increased local concentration of the protein. If the diffusion of the surface-bound protein is restricted to two, rather than three dimensions, the likelihood of possible encounters increases [31,33,36,37]. Indeed, the PrP(107-143) fibrils bind a small fraction of PrP(89-230) monomers (Appendix A; Figure A8). Therefore, it is possible that PrP(89-230) monomers can condense on the surface of the PrP(107-143) fibrils. However, PrP(107-143)-fibrils alone do not efficiently facilitate aggregation of monomeric PrP(89-230), suggesting that the local concentration of PrP(107-143) fibril-surface-bound PrP(89-230) monomers is very small and hence the low probability of new growth-competent nuclei formation. This could result due to a weak interaction between PrP(89-230) monomers PrP(107-143) fibrils [37]. If the monomer-fibril interaction strength is weak, protein monomers cover only a small fraction of the fibril surface, and the monomer adsorption and oligomer/nuclei formation on the fibril surface become the rate-limiting steps of the fibril self-replication reaction [37]. Almost identical secondary structure profiles of PrP(89-230) fibrils and the fibrils formed in the presence of PrP(107-143) fibrils (Figure 3 and Appendix A
Figure A1c) suggest that the rate of PrP(107-143) fibril self-replication reaction could be slow enough for spontaneous PrP(89-230) aggregation reaction to overcome it.

Second, existing aggregates can reduce the interfacial energy and lead to the formation of new nuclei. If protein monomers condense on the surface of fibrils, their contact with the solution is lower than when they are present in the bulk of the solution [33]. This way, if the interfacial energy is lower at the PrP(89-230) monomers-PrP(107-143) fibril contact area, then the free energy barrier of nuclei formation would be reduced [31,33].

Third, the interaction of the protein with the fibril surface may alter the nucleation process. For instance, the fibril-surface-bound monomented favorablrs can be oriey for the nucleation event to occur [33]. Moreover, protein adsorption on the surface of the fibril may induce structural rearrangements that would result in the appearance of accessible protein conformations that would be too high in energy in bulk solutions [33]. This could also lower the effective energy barrier of the nucleation reaction. In our case, we could assume that PrP(107-143) fibril-surface-bound monomers may acquire a conformation(s), which is not suitable for the formation of aggregation-prone nuclei on the PrP(107-143) fibril surface, or which can not be recruited by the PrP(107-143) fibril ends. Such monomers, however, could detach from the fibril surface into the bulk of the solution, where they could be subsequently recruited by the PrP(89-230) fibrils. This would facilitate the PrP(89-230) fibril self-replication reaction due to a lower effective energy barrier of nucleation/elongation. Moreover, it is likely that PrP(89-230) affinity for the fibrils formed by PrP(89-230) is substantially higher that that for fibrils formed by PrP(107-143), which means that if the PrP(89-230) fibrils would come into a close proximity with the PrP(107-143) fibrils, they could recruit PrP(89-230) monomers adsorbed onto the PrP(107-143) fibril surface. This would again facilitate self-replication reaction due to a lower effective energy barrier of the nucleation/elongation and due to increased local concentration of PrP(89-230) monomers. Additional experiments, during which we introduced PrP(107-143) fibrils at different time-points of PrP(89-230) self-replication reaction (Figure A3), revealed that the PrP(107-143) fibrils affect mainly the nucleation process. When the PrP(107-143) fibrils were added at the beginning of the self-replication reaction, the *t*${}_{lag}$ was ∼80 min shorter when compared to the *t*${}_{lag}$ of the sample where the addition of PrP(107-143) fibrils was delayed by 90 min. However, the *t*${}_{lag}$ difference between samples where PrP(107-143) fibrils were introduced after 90 min and 435 min delay is small. This suggests that the nucleation enhancing effect of PrP(107-143) fibrils is much stronger at later stages (i.e., at 435 min) of the aggregation reaction and this could be explained by the fact that the concentration of nuclei/fibril ends is much higher at this stage of the reaction. These results are in agreement with the data presented in Figure 1, where the synergistic effect of different PrP fibrils was most evident at the highest concentrations of PrP(89-230) fibrils. The opposite experiments, during which we introduced PrP(89-230) fibrils at different time-points of PrP(89-230) aggregation reaction (Figure A4) in the presence of PrP(107-143) fibrils were also performed. The experimentally determined *t*${}_{lag}$ values for the samples where PrP(89-230) fibrils were introduced at 180 and 360 min are equal (within the error margin) to the sum of the *t*${}_{lag}$ value determined for the sample where fibrils were introduced at the beginning of the reaction and the times of PrP(89-230) fibril introduction. This suggests that the presence of both PrP(107-143) and PrP(89-230) fibrils at the same time is important for the maximum acceleration of monomer nucleation.

In general, we can conclude that aggregates formed by PrP(107-147) are not able to seed aggregation reactions of monomeric PrP(89-230); however, they can facilitate the self-replication reaction of PrP(89-230) amyloid fibrils via several possible ways. A synergistic seeding effect of two different amyloid fibrils deserves more attention, as it may play an important role in amyloid propagation.

## 4. Materials and Methods

### 4.1. Measurements of Aggregation Kinetics

Mouse recombinant prion protein C-terminal fragment 89-230 (PrP(89-230)) was expressed in *E. coli* and purified according to the previously described protocol [34]. After purification, the protein was dialyzed against MilliQ water, and lyophilized. In addition to the PrP(89-230) sequence, the protein contains a 4-residue N-terminal extension (GSDP).

Synthetic mouse prion protein fragment 107-143 (PrP(107-143)) was produced as described previously [29].

To prepare PrP(89-230) fibrils, 5 mg of lyophilized PrP(89-230) was dissolved in 1 mL of 50 mM phosphate buffer (pH 6) containing 0.5 M GuHCl and filtered through 0.45 μm pore size syringe filter. The concentration of monomeric PrP(89-230) (M.W. = 16,555 Da, $\varepsilon_{280}$ = 27,515 M${}^{-1}$ cm${}^{-1}$) was determined by measuring UV-absorption at 280 nm using NanoDrop 2000 (Thermo Fisher Scientific, Wolsom, MA, USA). Subsequently, the PrP(89-230) solution was diluted to a final concentration of 0.5 mg/mL using 50 mM phosphate buffer (pH 6) containing 0.5 M GuHCl and incubated for 1 day at 60 ${}^{\circ }$C with 600 RPM shaking in a thermomixer MHR 23 (Ditabis, Pforzheim, Germany). For seeding experiments, PrP(89-230) fibrils were sonicated for 10 min on ice using Sonopuls 3100 (Bandelin, Berlin, Germany) ultrasonic homogenizer equipped with MS72 tip (using 20% power, cycles of 30 s/30 s sonication/rest, total energy applied to the sample per cycle ∼0.36 kJ). Right after the treatment, PrP(89-230) fibrils were mixed with 0.5 mg/mL PrP(89-230) solution in 0.5 M GuHCl in 50 mM phosphate buffer, pH 6, containing 50 μM ThT.

To prepare PrP(107-143) fibrils, 1 mg of synthetic PrP(107-143) was dissolved in 1 mL of MilliQ water and filtered through 0.45 μm pore size syringe filter. The concentration of the PrP(107-143) (M.W. = 3699 Da, $\varepsilon_{280}$ = 1490 M${}^{-1}$ cm${}^{-1}$) was determined by measuring UV-absorption at 280 nm using NanoDrop 2000 (Thermo Fisher Scientific, Wolsom, MA, USA). Subsequently, the peptide solution was diluted to a final concentration of 0.5 mg/mL using MilliQ water. Then 0.5 mg/mL PrP(107-143) solution was mixed with 40 mM sodium acetate, pH 3.7, supplemented with 280 mM NaCl in a 1:1 ratio, and incubated for 2 days at 25 ${}^{\circ }$C under quiescent conditions. Then, fibrils were centrifuged at 10,000× *g* for 30 min and re-suspended in 50 mM phosphate buffer (pH 6) containing 0.5 M GuHCl. The concentration of monomer remaining in the supernatant was determined by measuring UV absorption. For seeding experiments, fibrils were sonicated for 30 s on ice using Sonopuls 3100 (Bandelin, Berlin, Germany) ultrasonic homogenizer equipped with MS72 tip (using 20% of the power, total energy applied to the sample ∼0.36 kJ). Right after the treatment, fibrils were mixed with 0.5 mg/mL PrP(89-230) solution in 0.5 M GuHCl in 50 mM phosphate buffer, pH 6, containing 50 μM ThT.

For co-seeded aggregation experiments, sonicated PrP(89-230) and PrP(107-143) fibrils were co-added into 0.5 mg/mL PrP(89-230) solution in 0.5 M GuHCl in 50 mM phosphate buffer, pH 6, containing 50 μM ThT.

Aggregation kinetics at 60 ${}^{\circ }$C temperature were monitored by ThT fluorescence assay (excitation at 470 nm, emission at 510 nm) using Rotor-Gene Q (Qiagen, Hilden, Germany) real-time analyzer.

The ThT fluorescence intensity was normalized by dividing each data point by the maximum intensity of the curve. In the case of data where no change in ThT fluorescence intensity was evident, each data point was divided by the maximum intensity of the curve obtained in the presence of 5% of PrP seed. The aggregation lag time (t${}_{lag}$) values were calculated by applying a linear fit to the data points ranging from 40% to 60% of normalized intensity values and finding *x* value at y=0. We employed AmyloFit software [13] to analyze the kinetic data of the PrP(89-230) self-replication reaction in the absence and the presence of PrP(107-143) fibrils, but none of the available models gave an acceptable fit. The best fit was obtained using the Saturating Elongation and Secondary Nucleation model (Appendix A; Figure A9).

### 4.2. Atomic Force Microscopy (AFM)

AFM measurements were performed as described previously [38]. During the AFM imaging, to avoid large imaging force and keep consistency within independent samples, the constant regime of phase change not exceeding Δ20${}^{\circ }$ was maintained [39]. AFM images were flattened using Gwyddion software (Czech metrology institute, Jihlava, Czechia) as described previously [39].

### 4.3. Fourier-Transform Infrared (FTIR) Spectroscopy

FTIR measurements were performed as described previously [10].

## Figures and Tables

**Figure 1 ijms-21-07410-f001:**
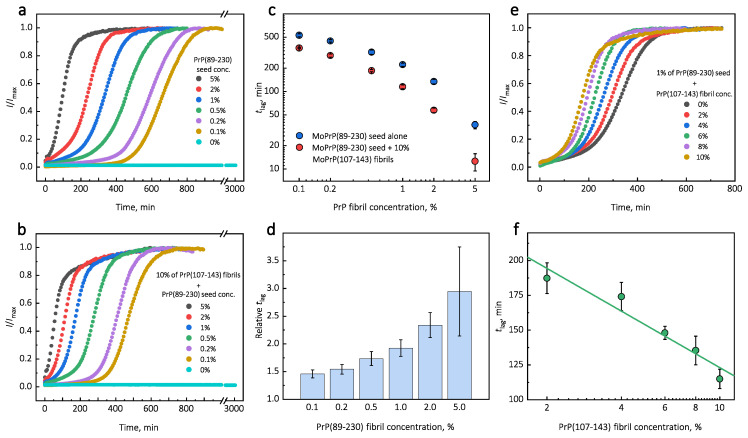
Representative curves of PrP-seed-induced aggregation reaction kinetics performed in the absence (**a**) or presence (**b**) of PrP(107-143) fibrils. Logarithmic plot of $t_{lag}$ values of seed-induced aggregation reaction performed in the absence and presence of 10% of PrP(107-143) fibrils (**c**). Relative $t_{lag}$ values (ratio between $t_{lag}$ in the absence and presence of 10% peptide fibrils) (**d**). Dependence of PrP(89-230)-seed-induced aggregation reaction kinetics on the initial concentration of PrP(107-143) fibrils (**e**,**f**). Error bars are standard deviations (*n* = 9).

**Figure 2 ijms-21-07410-f002:**
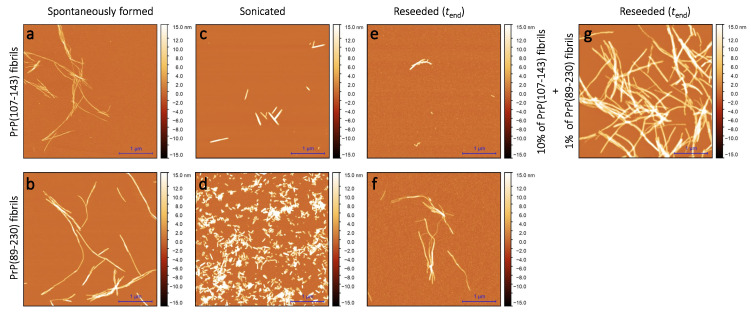
AFM images of PrP(107-143) and PrP(89-230) aggregates. Images of the spontaneously formed PrP(107-143) (**a**) and PrP(89-230) (**b**) aggregates. Images of sonicated PrP(107-143) (**c**) and PrP(89-230) (**d**) aggregates. Images of aggregates formed during PrP(107-143)-fibril-induced (**e**) or PrP(89-230)-seed-induced (**f**) aggregation reaction. Images of aggregates formed in the presence of both PrP(107-143)-fibrils and PrP(89-230)-seed (**g**).

**Figure 3 ijms-21-07410-f003:**
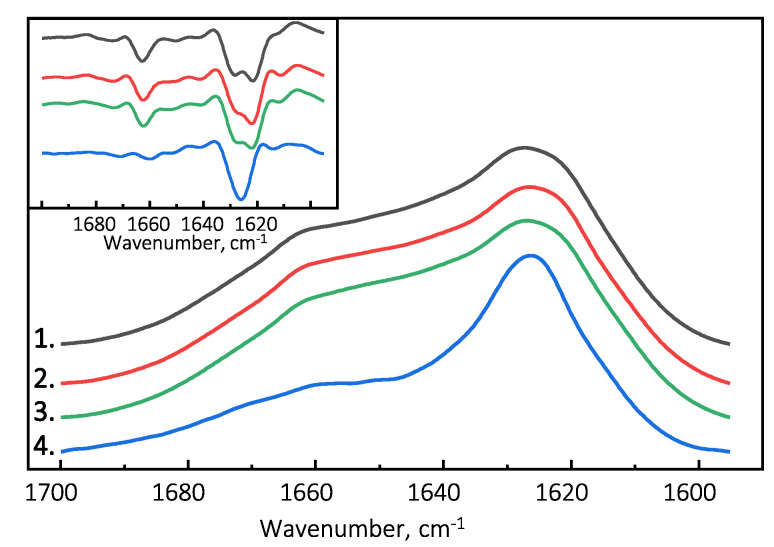
Absorbance and second derivative (inset) FTIR spectra of PrP(89-230) and PrP(107-143) fibrils. 1.—Spontaneously formed PrP(89-230) fibrils. 2.—Fibrils formed in the presence of 1% of PrP(89-230) fibrils. 3.—Fibrils formed in the presence of 10% of PrP(107-143) fibrils and 1% of PrP(89-230) fibrils. 4.—Spontaneously formed PrP(107-143) fibrils.

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
