# Peer review of "Self-Replication of Prion Protein Fragment 89-230 Amyloid Fibrils Accelerated by Prion Protein Fragment 107-143 Aggregates"

_ijms, 2020, doi:10.3390/ijms21197410_

Round 1

Reviewer 1 Report

Smirnovas and coworkers report on an interesting collaboration between short and long construct PrP fibrils on the seeding of long-construct PrP monomers. The data are of high quality (kinetics, AFM, FTIR) and competently interpreted.

Major issue:

Despite the above qualities, I am still left somewhat unsatisfied. The work is highly descriptive and I do not really get an understanding of the basis for the synergy – which surely should be the motivation for the study. The authors speculate eloquently on possible mechanisms in the Discussion but do not really follow up on it. Additional experiments such as the addition of short-construct/long-construct fibrils at different stages of the fibrillation process would shed light on this. Also the authors could consider some carefully designed studies with different concentrations of monomers in the presence of a fixed concentration of long and short fibrils to be analyzed using the programme Amylofit; this will shed more light on the dominance of different secondary processes (fragmentation/secondary nucleation) and the relative contributions of the different fibrils to this.

Minor points:

  1. “with high initial fibril concentrations resulting in an exponential shape curve and lower concentrations yielding sigmoidal shaped curves.” As far as I can see, all curves are sigmoidal, even at the highest seed concentration. Ultimately they might become exponential but this does not occur under the present seed concentration range.
  2. . Fig. 1c hows the rate constants approaching each other on a linear y-axis with higher fibril concentrations, yet Fig. 1d show them diverging. Panel 1c should have a log y-axis to make this more obvious. (When I digitize the data and do so myself, this becomes clearer).
  3. It is confusing and unnecessary that the authors have both supplementary information and an appendix – why not simply combine them?
  4. It is inappropriate to introduce new results in the discussion rather than keep them in the results section. Thus the first section of Discussion should be really at the beginning of Results.
  5. Figure A1 legend lacks reference to panel c, while panel d could be improved with additional labelling of e.g. PK concentration rather than simple lane numbers.

Author Response

Reviewer 1.

Smirnovas and coworkers report on an interesting collaboration between short and long construct PrP fibrils on the seeding of long-construct PrP monomers. The data are of high quality (kinetics, AFM, FTIR) and competently interpreted.

Major issue:

Despite the above qualities, I am still left somewhat unsatisfied. The work is highly descriptive and I do not really get an understanding of the basis for the synergy – which surely should be the motivation for the study. The authors speculate eloquently on possible mechanisms in the Discussion but do not really follow up on it. Additional experiments such as the addition of short-construct/long-construct fibrils at different stages of the fibrillation process would shed light on this. Also the authors could consider some carefully designed studies with different concentrations of monomers in the presence of a fixed concentration of long and short fibrils to be analyzed using the programme Amylofit; this will shed more light on the dominance of different secondary processes (fragmentation/secondary nucleation) and the relative contributions of the different fibrils to this.

We thank the reviewer for the suggestions, we have performed additional experiments during which we added PrP(107-143) fibrils at different time-points of PrP(89-230) self-replication reaction and discussed them in the manuscript (Lines 102-106; 192-201 and Appendix A Figure A3).

Indeed, Amylofit is a strong tool that enables to obtain mechanistic insights into amyloid aggregation prosess. Howewer, in our case the system is highly complex, as it consists from monomeric protein and two different type of aggregates, and we believe the mechanistic analysis of such complex system is worth a separate detailed study.

Minor points:

  1. “with high initial fibril concentrations resulting in an exponential shape curve and lower concentrations yielding sigmoidal shaped curves.” As far as I can see, all curves are sigmoidal, even at the highest seed concentration. Ultimately they might become exponential but this does not occur under the present seed concentration range.

We have modified the text (Lines 94-96):  

“PrP(89-230) aggregation reaction induced by addition of PrP(89-230) fibrils shows a typical fibril-concentration-dependent change in the lag phase (Figure 1a, c), with high initial fibril concentrations resulting in an exponential shape curve and lower concentrations yielding sigmoidal shaped curves.”

To :

“PrP(89-230)  aggregation  reaction  induced  by  addition  of  PrP(89-230)  fibrils  shows  a  typical fibril-concentration-dependent change in the lag phase (Figure 1a, c), yielding sigmoidal shaped curves.”

  1. 1c shows the rate constants approaching each other on a linear y-axis with higher fibril concentrations, yet Fig. 1d show them diverging. Panel 1c should have a log y-axis to make this more obvious. (When I digitize the data and do so myself, this becomes clearer).

We thank the reviewer for the suggestion, Fig. 1c y-axis was changed to log axis as suggested by the reviewer. The original Fig. 1c was moved to the Appendix A.

  1. It is confusing and unnecessary that the authors have both supplementary information and an appendix – why not simply combine them?

We have implemented supplementary information into the appendix. 

  1. It is inappropriate to introduce new results in the discussion rather than keep them in the results section. Thus the first section of Discussion should be really at the beginning of Results.

The first section of Discussion was moved to the beginning of the Results section as suggested by the reviewer.

  1. Figure A1 legend lacks reference to panel c, while panel d could be improved with additional labelling of e.g. PK concentration rather than simple lane numbers.

We have added a reference to panel c of Figure A1 legend and improved panel d by adding additional labelling and modifying text in the legend.

Reviewer 2 Report

The authors reported in their study that PrP fragment 107-143 accelerates the self-replication and aggregation of fragment 89-230. ß-sheet enriched aggregates were characterized by AFM and FTIR.

Altogether, the study is potentially interesting and well organized. Following points needs to be addressed further:

1)Fig. 1 lacks a negative arbitrary PrP sequence control, which underlines the crucial function of PrP 107-143 in PrP89-230 self replication. Why is PrP fragment 107-143 so important? How does it promote the aggregation of PrP89-230?

2) PrP fragments analysed in this study derived from mouse. The relevance/implication of the PrP107-143 for human prion diseases, such as the Creutzfeldt-Jakob disease needs to be discussed further.

3) Figure 2: How does PrP107-143 influence the morphology of PrP 89-230 fibrils? Does it also change the stability of PrP89-230 fibrils?

4) In Fig. A1: how do the authors explain the double bands in the WB after PK?

Round 2

Reviewer 1 Report

The authors have swiftly responded to my suggestions. All minor points have been addressed. However, as regards the major point:

1. The data in Fig. A3 are not described correctly in lines 102-106. The greatest effect (i.e. change) of t(lag) is obtained by adding fibrils at time zero, not at time 435 min where the effect is smallest. Also, I asked for the authors to add both short- and long-length fibrils for comparison and this still needs to be addressed.

2. I am not convinced that Amylofit should be dismissed so readily for a first analysis. It is straightforward enough to carry out those experiments (and the authors were impressively fast in responding to my suggestions anyway).

Reviewer 2 Report

All points has been addressed, therfore the manuscript can be now considered for publication.

Author Response

No response required.

Round 3

Reviewer 1 Report

The authors have now addressed my remaining points well. The only VERY final remark I would make is a friendly suggestion to add a few lines in the manuscript about the challenges/limitations of using Amylofit to fit the data. Given that this programme is set to become the standard way to analyze aggregation kinetics, it will benefit the readers to know that you have already "been there". Also it may encourage the Amylofit team (of which I am not a member) to extend their programme further.

Author Response

We added a couple of lines in methods section, as well as a figure of the fit in the appendix:

"We have employed AmyloFit software [13] to analyze the kinetic data of PrP(89-230) self-replication reaction in262the absence and presence of PrP(107-143) fibrils, but none of the available models gave an acceptable fit. The best fit was obtained using the Saturating Elongation and Secondary Nucleation model (Appendix A Figure A9)."